# Phylogeography and Biological Characterizations of H12 Influenza A Viruses

**DOI:** 10.3390/v14102251

**Published:** 2022-10-13

**Authors:** Zhimin Wan, Qiuqi Kan, Dongchang He, Zhehong Zhao, Jianxi Gong, Wenjie Jiang, Ting Tang, Yafeng Li, Quan Xie, Tuofan Li, Hongxia Shao, Aijian Qin, Jianqiang Ye

**Affiliations:** 1Key Laboratory of Jiangsu Preventive Veterinary Medicine, Key Laboratory for Avian Preventive Medicine, Ministry of Education, College of Veterinary Medicine, Yangzhou University, Yangzhou 225009, China; 2Institute of Agricultural Science and Technology Development, Yangzhou University, Yangzhou 225009, China; 3Jiangsu Co-Innovation Center for Prevention and Control of Important Animal Infectious Diseases and Zoonoses, Yangzhou 225009, China; 4Joint International Research Laboratory of Agriculture and Agri-Product Safety, The Ministry of Education of China, Yangzhou University, Yangzhou 225009, China

**Keywords:** H12 subtype IAV, wild birds, evolution, biological characterizations, phylodynamics

## Abstract

Influenza A virus (IAV) is widespread in wild bird reservoirs. Sixteen hemagglutinin subtypes are associated with wild waterfowl hosts; some subtypes are isolated infrequently, one of which is H12 IAV. In this study, we detected three H12 IAVs from *Anas*
*crecca* and *Anas formosa* in Poyang Lake, China, in 2018, one of which was isolated. Phylogenetic analysis revealed that the genome sequences of the three H12 viruses belonged to the Eurasian lineage, except for PA genes and one NP gene, which belonged to the North American lineage. The growth kinetics showed that the H12 isolate grew better in A549 than MDCK cells. Moreover, although the H12 isolate cannot efficiently replicate in BALB/c mice, it can bind to both *α*-2,6 sialic acid (SA) and *α*-2,SA-linked receptors. In addition, we examined the phylodynamics of H12 viruses by Bayesian phylogeographic analysis. The results show that two major transmission routes of H12 IAVs were from Asia to Oceania and from Europe to South America, and *Anas* and *Arenaria* genera were the major hosts of the viral transmission. Our findings help us better understand the evolution of H12 IAV and highlight the need for the continued surveillance of IAVs circulating in wild birds.

## 1. Introduction

Influenza A virus (IAV) belongs to the *Orthomyxoviridae* family and contains eight, negative-sense, single-stranded viral RNA segments [1]. Based on the antigenicity of the surface glycoproteins, IAVs have been grouped into 18 hemagglutinin (HA) and 11 neuraminidase (NA) subtypes [2,3]. With the exception of H17N10 and H18N11 detected in bats, all HA and NA subtypes have been detected in wild birds, especially in aquatic birds [3,4]. As the main natural reservoir of IAVs, wild birds not only maintain the circulation of IAVs in nature [4], but also spread viruses to poultry and mammals on their migration routes, posing a potential for the emergence of endemic or pandemic IAVs [5,6]. Therefore, the continued surveillance of IAVs in wild bird populations is necessary.

The evolution of IAVs in the natural reservoir is largely determined by the geographical separation and migratory patterns of wild birds. For this reason, IAVs are broadly separated into different lineages that correspond to geographic regions. IAVs of North American, Eurasian, and Australian lineages are well-defined due to long-term surveillance and the significant amount of IAV sequence information from these regions [4,7,8]. The Poyang Lake in Jiangxi Province is the largest freshwater lake in China, where wild birds and poultry share water, food, habitats, and other resources, increasing the transmission and reassortment of IAVs. Moreover, Poyang Lake is on the East Asian−Australasian Flyway of migratory birds, which plays a crucial role in the evolution of IAVs in wild birds. The surveillance of IAVs in Poyang Lake is crucial for understanding the ecology and evolution of IAVs in China.

H12 subtype IAV is also waterfowl-associated, but it is infrequently detected in nature; however, it can be reassorted with other subtypes [9,10,11]. H12 IAV was first detected in Canada in 1983. Limited research has been conducted on the phylodynamics and pathogenicity of H12 IAV. Their ecology and phylogeny remain largely unknown. In this study, three H12 IAVs were detected, and the genetic diversity was determined. One of them was successfully isolated in specific pathogen-free (SPF) eggs and its biological characterizations were evaluated.

## 2. Materials and Methods

### 2.1. Virus Detection and Isolation

A total of 981 cloacal samples were collected from wild birds in Poyang Lake (113°35′–118°29′ east longitude and 24°29′–30°05′ north latitude) in 2018. RNAs were extracted from these samples and cDNAs were synthesized by reverse transcription (RT) with the primer 5′-AGCAAAAGCAGG-3′ [12], and the M gene was amplified by PCR with the specific primers. The other seven segments of the positive samples were amplified by PCR with the specific primers [13], the amino acids were analyzed with DNAStar software (DNAStar7.1 USA), and all eight segments were sequenced by Sanger sequencing. The positive samples were inoculated into 10-day-old SPF embryonated chicken eggs for viral isolation. Allantoic fluid was harvested after 72 h and tested by hemagglutination (HA) assay.

### 2.2. Data Collection and Phylogenetic Analyses

The sequences of H12 IAVs in public databases (NCBI and GISAID) were collected by October 2021 and cleaned as described previously [14]. A total of 377 PB2, 382 PB1, 380 PA, 398 HA, 386 NP, 373 NA, 387 M, and 386 NS genes were collected. The sequences were aligned using MAFFT (v7.453) [15]. The duplicate and highly similar sequences were removed by BioAider (v1.423) [16]. The sequences with less than 90% completeness of the Open Reading Frame (ORF) were also removed. The general time-reversible (GTR) model with four gamma categories was selected as the Best-fit substitution model according to the Bayesian information criterion (BIC) value with ModelFinder in the PhyloSuite [17]. The maximum likelihood (ML) tree was constructed by RAxML-NG (V1.0.1) [18]. The reliability of the phylogenetic tree was estimated 1000 times using bootstraps. The ML tree was inspected by the root-to-tip genetic distance against sampling data using TreeTime (v0.8.1), and incorrect sampling dates or abnormal root-to-tip distance were removed from alignments [19].

### 2.3. Evolutionary and Phylodynamics Analyses

The time-resolved tree was inferred by the Markov chain Monte Carlo (MCMC) framework applied in Bayesian Evolutionary Analysis Sampling Trees (BEAST, v1.10.4) [20]. The GTR + G4 nucleotide substitution model was chosen based on the results in ModelFinder. An uncorrelated relaxed molecular clock model and Bayesian SkyGrid coalescent tree prior were chosen [21]. The length of the MCMC chain was set to 200 million generations, and trees were collected every 20,000 steps. TreeAnnotator analyzed the maximum clade credibility (MCC) tree with common ancestor heights following the burn-in of the first 10%. The phylogenetic tree was plotted and visualized using ggtree and ggtreeExtra in R (v4.1.0).

To gain insight into the host and spatial dynamics of H12 viruses, Bayesian phylogeographic analysis was performed using BEAST. Each HA gene sequence was assigned two discrete traits (geographical location and host species). The geography was divided into North America, South America, Europe, Asia, and Oceania. These five continents were chosen and coded as discrete states. The spatial diffusions and host transmissions were inferred as asymmetric using Bayesian stochastic search variable selection (BSSVS) [22,23]. SpreaD3 (v.0.9.7) was used to visualize the spreading process over time and calculate the Bayes factor (BF) support for each route [24]. Significant migration routes were determined based on the established criteria (BF > 3 and the posterior probability > 0.5). Values of 3 ≤ BF < 10, 10 ≤ BF < 100, 100 ≤ BF < 1000, and BF ≥ 1000 indicated substantial, strong, very strong, and decisive support, respectively.

### 2.4. In Vitro Growth Kinetics

The growth kinetics were determined in MDCK and A549 cells. Briefly, the confluent cells were infected with H12 isolate at a multiplicity of infection (MOI) of 0.01, maintained in Opti-MEM media containing 1 µg/mL TPCK-trypsin and incubated at 37 °C. The tissue culture supernatants were harvested at 6, 12, 24, 48, and 72 h post-infection (hpi), and then virus titers were measured by TCID_50_ in MDCK or A549 cells. The experiments were performed in triplicate.

### 2.5. Solid-Phase Binding Assay

The receptor-binding property of the H12 isolate was analyzed by a solid-phase direct-binding assay [25]. Briefly, the Pierce streptavidin high-binding capacity 96-well plates (Thermo, Rockford, USA) were coated serially in 2-fold diluted Neu5Ac*α*2-3Galb1-4GlcNAcb-PAA-biotin and Neu5Ac*α*2-6G*α*lb1-4GlcNAcb-PAA-biotin (Glycotech, Gaithersburg, MD, USA) and incubated overnight at 4 °C. After being blocked with 5% skim milk in PBST, the plates were incubated at 4 °C overnight with 64 HAUs of H12 isolate. After washing 3 times with PBST, the plates were incubated with serum against the H12 virus at 4 °C for 2 h. The plates were washed 3 times with PBST and incubated with a horseradish peroxidase (HRP)-conjugated, anti-mouse IgG antibody for 2 h at 4 °C and then washed 3 times with PBST. Then, 100 µL TMB solution was added to each well and incubated for 10 min. The reaction was stopped by adding 2 M H_2_SO_4_, and OD450 values were measured.

### 2.6. Animal Study

Twelve five-week-old BALB/c mice were anesthetized with isoflurane and infected with the H12 IAV at 10^5^ EID_50_ in 25 μL of PBS by intranasal inoculation. At 3 and 5 days post-infection (dpi), three mice from each group were humanely euthanized, and the lungs were collected to determine virus titers in 10-day-old SPF eggs. The remaining 6 mice in each group were monitored daily for body weight loss and clinical signs. The mice with a body weight loss of more than 25% were anesthetized and euthanized.

### 2.7. Statistical Analysis

All the results are presented as means ± standard deviations. The statistical analysis in this study was performed with a Student t-test using Graphpad 5 software (San Diego, CA, USA). A *p* value of below 0.05 was considered significant. *, **, and *** indicate *p* values of less than 0.05, 0.01, and 0.001, respectively.

### 2.8. Ethics Statement

The animal study was performed in accordance with institutional animal guidelines and the protocol approved by the Animal Care Committee at Yangzhou University, China. The ethical permission code (no. NSFC2020-SYXY-10, licensed on 23 March 2020) was provided by the Animal Ethics Committee of Yangzhou University.

## 3. Results

### 3.1. Spatial and Temporal Distribution of H12 IAVs

A total of five H6, one H9N2, and three H12 subtype IAVs were identified from the 981 cloacal samples collected from wild birds in Poyang Lake in 2018. Three H12 viruses were designated as A/*Anas crecca*/Jiangxi/2018WB0135/2018 (H12N2) (A. crecca/135, H12N2), A/*Anas formosa*/Jiangxi/2018WB0354/2018 (H12N1) (A. formosa/354, H12N1), and A/*Anas crecca*/Jiangxi/2018WB0421/2018 (H12N1) (A. crecca/421, H12N1). However, only A. formosa/354 was recovered from inoculating 10-day-old SPF embryonated chicken eggs. Then, we systematically analyzed the spatial and temporal distribution of H12 IAVs circulating in birds. As shown in Figure 1A, the most prevalent subtypes of H12 IAVs were H12N5 (215/396, 54.3%) and H12N4 (46/396, 11.6%); so far, only 12 H12 viruses have been reported in China. They were mainly detected from *Anas* (8/12). The common teal or Eurasian green-winged teal (6/12) is a common and widespread duck that breeds in temperate Euro-Siberia and migrates south in winter [26], and H12 viruses detected in China have been reassorted among IAVs from domestic poultry and wild birds [27]. Host analysis showed that the dominant hosts of H12 viruses detected in North America were *Anas* and *Arenaria*, while in Europe, the major host of H12 viruses is *Anas*. In contrast, the hosts of H12 viruses in Asia are relatively diverse (Figure 1A, C). The hosts of the three H12 IAVs detected in this study all belonged to *Anas*. Figure 1A,B show that most H12 viruses were detected in North America (83.8%), especially in the last two decades. Moreover, H12 viruses detected in China are mainly concentrated in the most recent decade (Figure 1B).

### 3.2. Phylogenetic Analyses of H12 IAVs

To better understand the evolution of H12 IAV and the three H12 viruses detected in this study, we collected all effective sequences of H12 IAVs (377 PB2, 382 PB1, 380 PA, 398 HA, 386 NP, 373 NA, 387 M, and 386 NS genes) from GISAID and NCBI for analysis. The ML phylogenetic tree was used to plot root-to-tip divergence against the sampling date and revealed that the H12 IAV sampling date was sufficient to correct the molecular clock in the subsequent time-resolved tree analysis (Figure 2A, R^2^ = 0.66). The MCC tree of HA showed that the HA genes of all H12 viruses, including the three H12 IAVs identified in this study, diverged into two major geographical lineages—Eurasian and North American—and the three H12 viruses detected here belonged to the Eurasian lineage (Figure 2B). Moreover, as shown in Figure 2B, the H12 viruses of North American lineage were mainly circulated in the wild bird population from North America. However, some H12 strains of North American lineage were detected in South America, Europe, and Asia, which were probably introduced through intercontinental bird migration.

To better understand the genetic diversity of the H12 viruses, the internal genes were analyzed by ML trees. The results show that all internal genes of the H12 viruses diverged into the Eurasian lineage and North American lineage (Figure 3). Similarly, the majority of the internal genes of these three H12 IAVs were genetically close to those H12 viruses circulating in Eurasia. Of interest is that the PA genes of all three H12 viruses and NP of A. formosa/354 were genetically close to H12 IAVs of North America (Figure 3). 

### 3.3. Phylodynamics of H12 IAVs

The Bayesian phylogeographic analysis was used to elucidate the host and spatial transmission patterns of H12 IAVs. Fifteen different host transmission links in the diffusion of H12 IAVs were analyzed (Appendix A). As described in Appendix A, six host transmission links are decisively supported, including from *Anas* to *Arenaria*, *Uria,* and *Calidri*s, and from *Arenaria* to *Cygnus*, *Mareca*, and *Leucophaeus* (BF > 1000); four host transmission links are very strongly supported, including from *Larus* to *Anas*, *Arenaria* to *Sibirionetta*, environment to *Somateria,* and from *Larus* to *Shorebird* (100 < BF < 1000); and the other five host transmission links are strongly supported (10 < BF < 100). Overall, these results suggest that *Anas* and *Arenaria* were major hosts of H12 IAV and then it diffused to other wild bird species (Figure 4A). Moreover, we analyzed the spatial transmission patterns in five geographic regions of the world (Asia, Europe, North America, South America, and Oceania). There are six migration links in the diffusion of H12 viruses (Figure 4B and Appendix A). Two transmission routes are decisively supported, including migration from Asia to Oceania (BF = 29,353.79) and from Europe to South America (BF = 4889.581). The routes from Europe to Oceania, from North America to Europe, and from North America to Asia are strongly supported (10 ≤ BF < 100) (Appendix A). Overall, these data suggest that Asia and Europe play an important role in seeding H12 viruses throughout the world.

### 3.4. Molecular Characterization of Viral Genes of H12 IAVs

Molecular characterization analysis indicated that the three H12 IAVs carry the same HA cleavage site motif PQAQDR/GL, typical for low pathogenicity (Table 1). Residues 226Q and 228G (in H3 numbering) (Table 1) in HA indicated that these three strains are likely to bind to *α*-2,3 SA [28]. The presence of 119E, 274H, 276E, 277E, and 292R in NA (N2 numbering) [29,30] and 31S in M2 [31] suggested the three strains were highly sensitive to antivirals. Other common molecular markers in the internal genes also were analyzed. E627K and D701N in PB2 [32], and V63I in PA [33] were not found in these three isolates, suggesting that the three H12 viruses have low pathogenicity in mice; however, L473V in PB1 [34], N30D and T215A in M1, and I106M in NS1 were identified, which have been shown to be able to increase the pathogenicity of the influenza virus in mice [35]. Taken together, the molecular characterization suggested that these three H12 IAVs have low pathogenicity and are highly sensitive to antivirals.

### 3.5. The Growth Characteristic of A. Formosa/354 In Vitro

To better understand the biological characteristics of the H12 isolate, first, A. formosa/354 virus was amplified and titrated in MDCK, A549 cells, and 10-day-old embryonated chicken eggs. The titer of A. formosa/354 was 10^6.199^ TCID_50_/_mL_ in MDCK cells and 10^6.366^ TCID_50_/_mL_ in A549 cells, respectively (Figure 5). Compared with MDCK and A549 cells, A. formosa/354 H12 IAV grows better in SPF chicken eggs with the titer of 10^8.699^ EID_50_/_mL_ (Figure 5A). Then, we examined the growth kinetics of A. formosa/354 in MDCK and A549 cells. The confluent monolayers of MDCK and A549 cells were infected with A. formosa/354 at an MOI of 0.01. The growth kinetics results show that A. formosa/354 virus grew to higher titers in A549 cells than in MDCK cells from 6 hpi, and this trend was maintained throughout the whole time course. The viral titers peaked at 24 hpi in both MDCK cells (10^4^ TCID_50_/_mL_) and A549 cells (10^6^ TCID_50_/_mL_) (Figure 5).

### 3.6. A. Formosa/354 H12 IAV Bind to Both α-2,3 SA and α-2,6 SA

The receptor-binding specificity of HA is a key determinant of the viral host range. A. formosa/354 was tested with Neu5Ac*α*2-3Galb1-4GlcNAcb (3′SLN) and Neu5Ac*α*2-6Galb1-4GlcNAcb (6′SLN) for receptor-binding preference. A/chicken/Jiangsu/X1/2004 (X1, H9N2) and A/Puerto Rico/8/1934 (PR8, H1N1) were included as controls for *α*-2,3 sialic acid (SA) and *α*-2,6 SA, respectively. As shown in Figure 6, A. formosa/354 isolate could bind both *α*-2,3 SA and *α*-2,6 SA (Figure 6).

### 3.7. A. Formosa/354 H12 IAV Could Not Efficiently Replicate in Mice

To further investigate the infection potential of A. formosa/354 IAV to infect mammals, 5-week-old BALB/c mice were infected intranasally with 10^5^ EID_50_ of A. formosa/354 IAV. Clinical signs of disease, body weight changes, and mortality were recorded daily. On 3 and 5 dpi, three mice from each group were euthanized and the lungs were collected to determine the viral titer in SPF eggs. As shown in Figure 7, the infected mice did not show any weight loss or any clinical signs of disease. Not surprisingly, the lung samples collected from 3 or 5 dpi mice had no detected virus (data not shown). These results show that A. formosa/354 H12 IAV could not efficiently replicate in BALB/c mice.

## 4. Discussion

Wild aquatic birds are the natural host for IAVs. H1–H16 subtypes are present in wild birds; however, some subtypes, e.g., H8, H12, H14, and H15, have rarely been detected [9,10,11,36,37]. In this study, three H12 subtype IAVs were detected in wild birds in Poyang Lake. Then, we focused on the evolution of H12 IAVs circulating in wild birds. The results show that H12 IAVs diverged into the Eurasian and North American lineages. As shown in Figure 1B, most H12 viruses were detected in North America. The majority of H12 viruses detected in North America belong to the North American lineage, which contains a large amount of genetic diversity (H12N1–H12N9) circulating in North America. Occasionally, those H12 viruses spilled over to Eurasia and South America, which were probably introduced through intercontinental bird migration (Figure 1 and Figure 2). The three H12 viruses detected in this study belonged to the Eurasian lineage (Figure 2). Moreover, PA genes of the three H12 IAVs and NP gene of A. formosa/354 were close to those of H12 IAVs of the North American lineage (Figure 3). These proved that H12 IAVs occurred in the reassortant event between Eurasia and North America. Since our results were analyzed based on the deposited data in NCBI and GISAID, the sampling bias could have affected the robustness of the phylodynamic results.

The H8, H9, H11, and H12 subtypes of IAV have a strong NA bias, whereby the frequently detected HA-NA subtype combination includes H8N4, H9N2, H11N2, H11N9, and H12N5 [9]. In this study, we also showed that most HA-NA subtype combinations of H12 IAV were H12N5 (54.3%), and H12N5 subtype IAVs provide HA genes for other H12 subtype IAVs (Figure 1 and Figure 3). Of interest is that the subtypes of H12 viruses circulating in China are H12N2 (5/12), H12N1 (2/12), and H12N8 (2/12). We need to continuously monitor the HA-NA subtype combinations of H12 IAV in China.

IAVs can cross species through reassortment and point mutations to adapt in a new host. The surveillance of IAVs in wild birds is therefore important to understand their evolution. Poyang Lake, one of the most important wetlands on the East Australasian Flyway of migratory birds, has over 10 million birds congregating each winter. Previous reports indicated that multiple IAV subtypes were detected in Poyang Lake, including H3, H4, H5, H6, H9, and H10 [38,39,40,41,42]. In this study, we showed that the hosts of *Anas* and *Arenaria*, and two migration routes of from Asia to Oceania and Europe to South America, play important roles in the diffusion of H12 IAV in wild birds (Figure 2). The East Asian–Australasian Flyway is probably responsible for H12 spreading from Asia to Oceania. For the migration route from Europe to South America, H12 probably spread through an unknown way or overlapped flyway. However, the H12 viruses from South America were clustered with those from America in the MCC tree (Figure 2B), indicating that the H12 spread in South America also probably came from North America. Since the H12 virus is difficult to isolate, H12 data might be missing, and the sampling bias could affect the robustness of the results.

Within waterfowl, some subtypes are detected at high frequencies every year. However, some subtypes were rarely detected, including H8, H12, H14, and H15. The biological characterizations of these rare IAVs are poorly understood. In this study, we only isolated one strain of H12 IAV (A. formosa/354) in the three positive samples, possibly due to the poor adaptability of these H12 viruses to chicken embryos. The biological characterizations of A. formosa/354 were determined. Although the H12 isolate cannot efficiently replicate in BALB/c mice, A. formosa/354 had distinct replication abilities in eggs, MDCK, and A549 cells (Figure 5). Interestingly, although A. formosa/354 does not contain amino acids that enhance the binding to human receptors (Table 1), it can bind to both avian-like and human-like receptors (Figure 6). The molecular basis for this needs to be further elucidated.

In summary, this study suggests that the emergence of genetically diverse H12 viruses is associated with wild bird migration. We propose that continued surveillance to monitor the diversity of H12 IAVs in wild birds is necessary to increase our understanding of the natural history of IAVs.

## Figures and Tables

**Figure 1 viruses-14-02251-f001:**
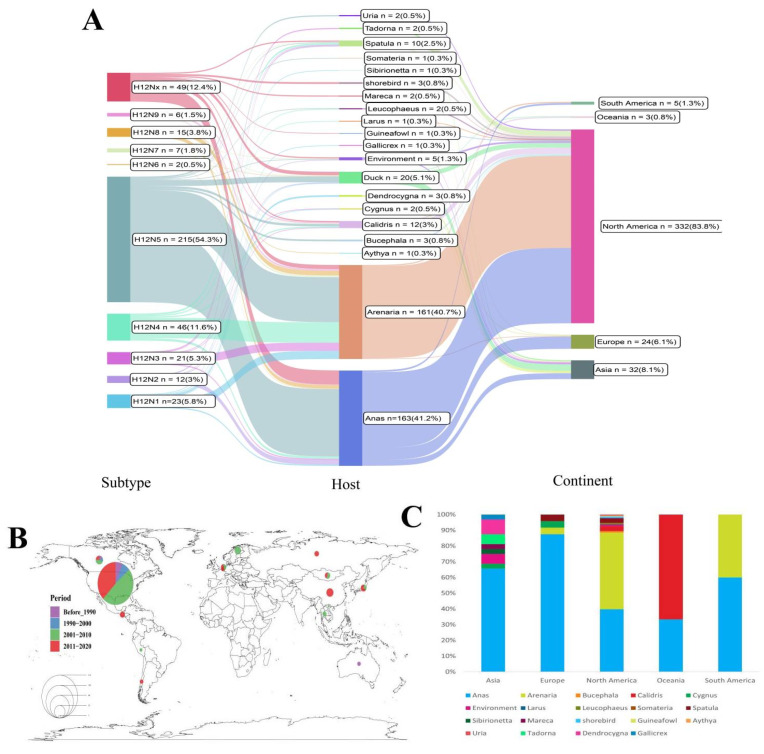
The detailed information of H12 IAVs. (**A**) The map of the subtypes, hosts (including genus/type of birds or environment), and continents of all H12 IAVs. (**B**) The spatial and temporal distribution of H12 IAVs. The pie chart shows the composition of the different periods, and the circle size represents the number of H12 strains. (**C**) The host species of H12 IAVs detected in 5 major regions around the world.

**Figure 2 viruses-14-02251-f002:**
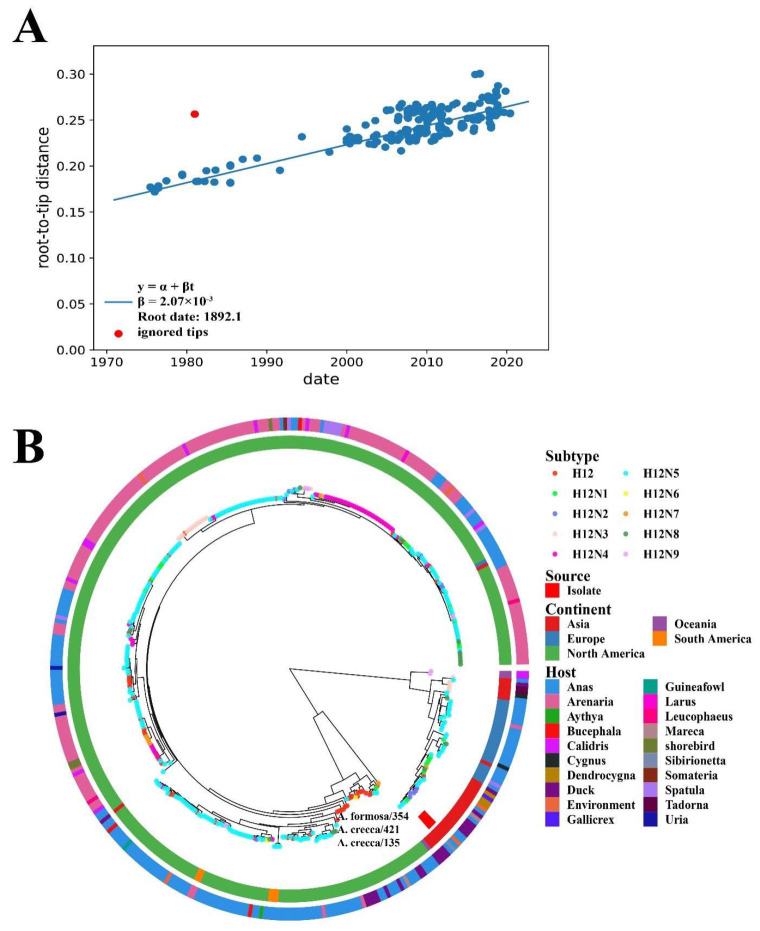
Phylogenetic analysis of HA genes of H12 IAVs. (**A**) The temporal signal was estimated. The regression line (slope in blue) represents the average evolutionary rate and is displayed as root-to-tip distance. (**B**) HA genes of the three H12 viruses detected in wild birds in the study and 398 HA genes of H12 viruses obtained from the GISAID and NCBI were analyzed. The MCC tree was analyzed using TreeAnnotator with common ancestor heights following the burn-in of the first 10%. The phylogenetic tree was plotted and visualized using ggtree and ggtreeExtra in R.

**Figure 3 viruses-14-02251-f003:**
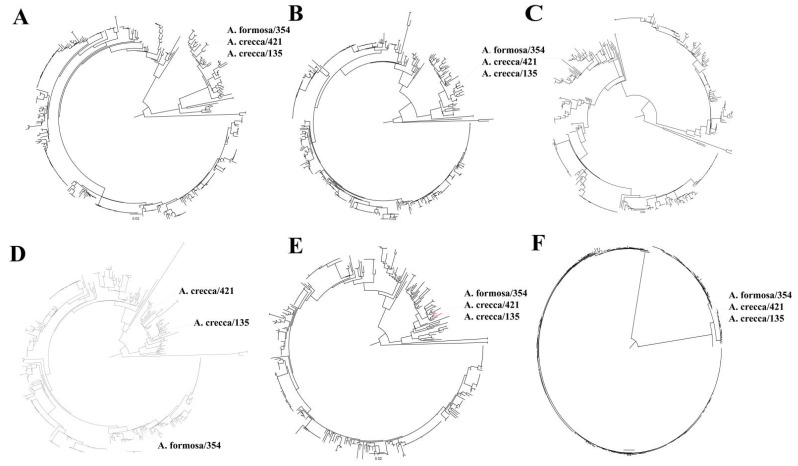
Phylogenetic analysis of internal genes of H12 IAVs. (**A**) PB2 genes, (**B**) PB1 genes, (**C**) PA genes, (**D**) NP genes, (**E**) M genes, and (**F**) NS genes of the three H12 viruses detected in wild birds in this study and obtained from GISAID and NCBI were analyzed. The three H12 isolates are marked in red. The ML tree was constructed by RAxML-NG. The reliability of the phylogenetic tree was estimated with 1000 bootstraps.

**Figure 4 viruses-14-02251-f004:**
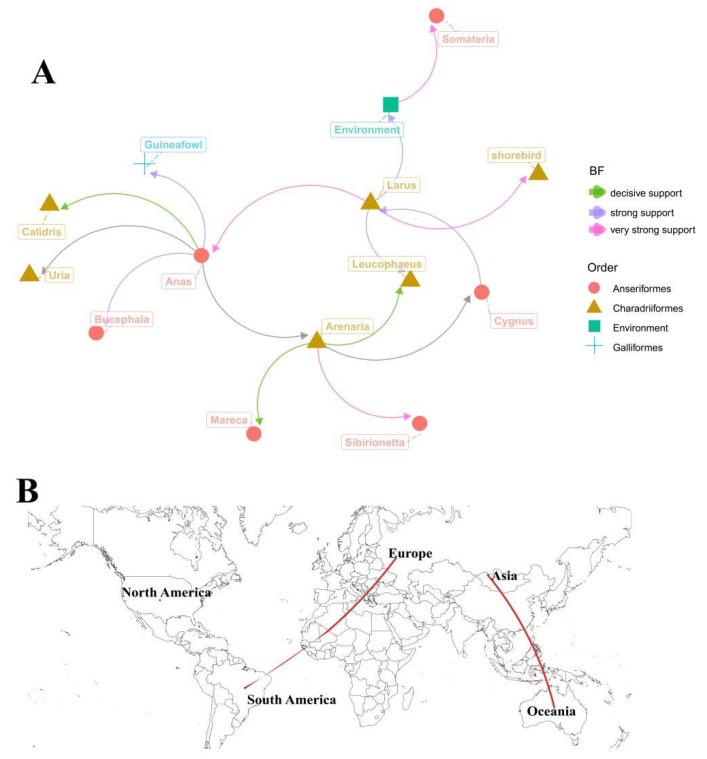
Host and spatial diffusion of H12 IAVs. Major hosts (**A**) and migration routes (**B**) of the viral transmission were determined based on the established criteria (BF > 3 and the posterior probability >0.5). Values of 3 ≤ BF < 10, 10 ≤ BF < 100, 100 ≤ BF < 1000, and BF ≥ 1000 indicated substantial, strong, very strong, and decisive support, respectively.

**Figure 5 viruses-14-02251-f005:**
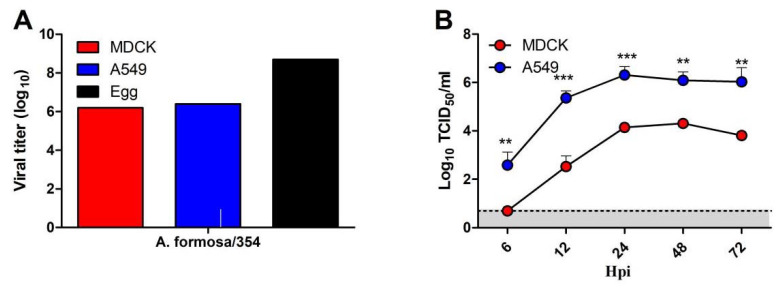
Viral titers and growth kinetics of H12 virus. (**A**) H12 virus was titrated in MDCK, A549 cells, and SPF eggs. (**B**) Viral growth kinetics was determined for H12 isolate in MDCK and A549 cells after inoculation at an MOI of 0.01. Supernatant samples were collected at 6, 12, 24, 48, and 72 hpi, and viral titers were measured in MDCK or A549 cells. ** and *** indicate *p* values of less than 0.05, 0.01, and 0.001, respectively.

**Figure 6 viruses-14-02251-f006:**
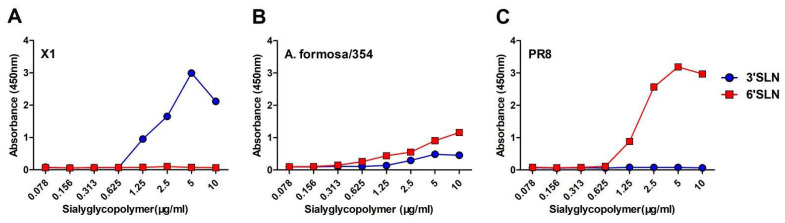
Receptor-binding properties of the H12 isolate. (**B**) The binding of the H12 isolate with sialic acid was determined using various concentrations of sialic acid conjugated to biotinylated sialylglycopolymers (3′SLN and 6′SLN) via direct solid-phase binding assays. (**A**) X1 and (**C**) PR8 were the controls for *α*-2,3 sialic acid (SA) and *α*-2,6 SA.

**Figure 7 viruses-14-02251-f007:**
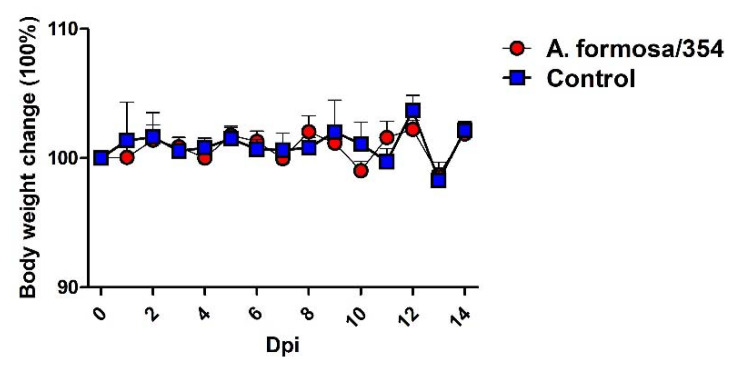
Body weight change in mice infected with H12 isolate from wild birds, Poyang Lake. Five-week-old BALB/c mice were infected with 10^5^ EID_50_. The body weight of mice was observed over 14 dpi.

**Table 1 viruses-14-02251-t001:** Specific amino acid residues analysis of three H12 IAVs identified from wild birds in Poyang Lake.

Protein	Mutation	Virus Strain			Function
		A. Crecca/135	A. Formosa/354	A. Crecca/421	
HA	Q226L ^a^	Q	Q	Q	Enhanced binding to human-type receptor [28]
	G228S	G	G	G	
NA	E119A ^b^	E	E	E	Reduced susceptibility to oseltamivir [29,30]
	H274Y	H	H	H	
	E276D	E	E	E	
	E277Q	E	E	E	
	R292K	R	R	R	
PB2	E627K	E	E	E	Increased polymerase activity in mammalian cell line [32]
	D701N	L	L	L	
PB1	L473V	V	V	V	Increased polymerase activity in mammalian cell line [34]
PA	V63I	V	V	V	Increased polymerase activity in mammalian cell line [33]
M1	N30D	D	D	D	Increased virulence in mice [35]
	T215A	A	A	A	
M2	S31N	S	S	S	Increased resistance to amantadine and rimantadine [31]
NS1	P42S	A	A	A	Increased pathogenesis in mice [35]
	I106M	M	M	M	

^a^ H3 numbering, ^b^ N2 numbering.

## Data Availability

Not applicable.

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
