# Peer review of "Phylogeography and Biological Characterizations of H12 Influenza A Viruses"

_viruses, 2022, doi:10.3390/v14102251_

Round 1

Reviewer 1 Report

Brief summary

In this article the Authors analyzed three novel H12 influenza A viruses (IAVs) detected and isolated (1/3) from Anas spp. in Poyang Lake of China in 2018. Phylogenetic analysis showed that the genome sequences of the three H12 viruses belonged to Eurasian lineage, except for PA genes and one of NP gene, which belonged to North American lineage. The growth kinetics showed that the H12 isolate grew better in A549 than MDCK cells. This H12 isolate cannot efficiently replicate in BALB/c mice, but it can bind to both α-2,6 sialic acid (SA) and α-2,3 SA.

The phylodynamics of H12 viruses, examined by Bayesian phylogeographic analysis, showed these spatial and transmission patterns: two major transmission routes of H12 IAVs were from Asia to Oceania and from Europe to South America, and Anas and Arenaria genera represented the major hosts of the viral transmission.

According to the Authors’ conclusions, the data obtained help to better understand the evolution of H12 IAV and emphasize the need for IAV surveillance activities in wild birds.

Broad comments

This article provides an in-depth scientific analysis aimed to better understand the evolution of H12 IAV in wild birds. Authors performed virus detection and isolation followed by molecular characterization, in vitro growth kinetics and animal study experiments as well as phylogenetic, evolutionary and phylodynamic analyses.

However, in my opinion, the manuscript needs some substantial revisions and minor modifications aimed to improve its clarity.

Please, see “Specific comments” for details.

Specific comments

Abstract:

  • Pag. 1_19: “..... is ubiquitous in wild birds.”. I suggest to modify this sentence by specifying that IAV is widespread in the wild bird reservoir.
  • Pag. 1_19-20: I suggest to integrate the sentence “Although hemagglutinin subtypes of H1-H16 are …. as follows “Sixteen hemagglutinin subtypes (from H1 to H16) are ….
  • Pag. 1_20: “very infrequently”: you could use the adverb “infrequently” only; alternatively, could you specify in which continents H12 viruses are more rarely detected?
  • Pag. 1_20: I suggest to complete the genus “Anas ….” with the duck species found to be H12 positive “Anas crecca and A. Formosa ….”. Please, see also the comment to the results on Pag. 3_141.
  • Pag. 1_26: I suggest to integrate “…. α-2,6 sialic acid (SA) and α-2,3 SA.” with “…. α-2,6 sialic acid (SA) and α-2,3 SA-linked receptors.”
  • Pag. 1_27-29: The results showed that two major transmission routes of H12 IAVs were from Asia to Oceania and from Europe to South America, and Anas and Arenaria were the major hosts of the viral transmission.” Please see comments on Pag. 6_205-207.
  •  Pag. 1_29: I suggest to integrate “…., and Anas and Arenaria were…..” as follows“…., and  Anas and Arenaria genera were …..”

Introduction:

  • Pag. 1_40: “…. migratory waterfowl …..”. Did you mean Anseriformes birds, or more generally " aquatic birds" (larger group of birds associated with water habitats)? Please, specify.
  • Pag. 2_56: “H12 subtype IAV is also waterfowl-associated, but it is infrequently detected in nature; however, it can be reassort with other subtypes [9]. In addition to [9] , I suggest adding further pertinent references, such as: i) Tang L, Tang W, Ming L, Gu J, Qian K, Li X, Wang T, He G. Characterization of Avian Influenza Virus H10-H12 Subtypes Isolated from Wild Birds in Shanghai, China from 2016 to 2019. Viruses. 2020 Sep 25;12(10):1085. doi: 10.3390/v12101085; ii) Sharshov K, Mine J, Sobolev I, Kurskaya O, Dubovitskiy N, Kabilov M, Alikina T, Nakayama M, Tsunekuni R, Derko A, Prokopyeva E, Alekseev A, Shchelkanov M, Druzyaka A, Gadzhiev A, Uchida Y, Shestopalov A, Saito T. Characterization and Phylodynamics of Reassortant H12Nx Viruses in Northern Eurasia. Microorganisms. 2019 Dec 3;7(12):643. doi: 10.3390/microorganisms7120643.

Materials and Methods:

  • Pag. 2_62: methods related to the Amino acid analysis of the Three H12 viruses are lacking in this section.
  • Pag. 2_64: A total of 981 cloacal samples were collected from wild birds in Poyang Lake in 2018.. If possible, could you add some information about the sampling methods and study area (e.g., geographic coordinates, lake size), the samples collected (numbers of wild birds, grouped at least by taxonomic order), and the study period (e.g. months of sampling)?
  • Pag. 2_73: The sequences of H12 IAVs in public databases (NCBI and GISAID) were collected ….. Please, specify the download period. I suggest adding a sentence about the number of sequences in the obtained database. Moreover, does this database also include data from domestic birds? if so, could you please quantify this sample?
  • Pag. 3_106: “Vitro growth kinetics”. I suggest to replace with “In vitro growth kinetics”.

Results:

  • Pag.3_141: Due to recent updates of scientific and common names of wild anatids, I suggest specifying the scientific names of Eurasian teal (Anas crecca) https://birdsoftheworld.org/bow/species/gnwtea/cur/introduction and Baikal teal (Anas formosa, more recently classified as Sibirionetta formosa).https://birdsoftheworld.org/bow/species/baitea/cur/introduction.
  • Pag.4_147-149: however, the dominant subtypes of H12 vruses circulating in China were H12N2 (5/12), H12N1 (2/12) and H12N8 (2/12) (data not shown)”. In general, the data obtained from China should be further discussed (e.g., origin of H12 viruses and interactions between wild and domestic birds). Please, see also: Liu J, Zhou L, Luo Z, Zou Y, Lv J, Wang T. Genomic characteristics and phylogenetic analysis of the first H12N2 influenza A virus identified from wild birds, China. Acta Virol. 2020;64(1):104-110. doi: 10.4149/av_2020_113.
  • Pag.4_149: Host genus analysis showed that dominant hosts ……”.  Actually, this analysis mainly includes hosts genera, but also some different IAV origins are reported (shorebirds, guineafowl, environment, ducks). Please, specify this.
  • Pag. 4_152-160: Figure 1 is very interesting and useful. However, the small font size and poor resolution hinder the consultation of the legend. I believe that figures 1A, 1B, and 1C should be enlarged and layout differently. Also, I wonder if your three H12 viruses are included in Figure 1. Finally, I would like to point out that the genus Sibirionetta included in the Figure 1 dataset is (very likely) a Baikal teal.
  • Pag. 4_163-164: “(377 PB2, 382 PB1, 380 PA, 398 163 HA, 386 NP, 373 NA, 387 M and 386 NS)”. Please integrate as follows “(377 PB2, 382 PB1, 380 PA, 398 163 HA, 386 NP, 373 NA, 387 M and 386 NS genes).
  • Pp. 5 and 6: As pointed out for Figure 1, the legend of Figure 2B and, in general, Figures 3A through 3F are also difficult to understand.
  • Pag. 6_205-207: “Two transmission routes are decisively supports, including migration from Asia to Oceania (BF=29353.79) and Europe to South America (BF=4889.581).” Although statistically supported, the transmission pattern of H12 virus from Europe to South America, also shown in Figure 4B, does not find an explanation in the birds' natural migratory routes. This result should be interpreted/discussed.
  • Pag. 7_ Figure 1A: Please check the color of the arrows in both the figure and the legend.
  • Pp. 7 and 8_Table 1: Please check virus strains for E627K and L473V mutations.
  • Pag. 8: 235-236: Please, amend “106.366 235 TCID50/ml”.
  • Pag. 8_236: Please replace “Figure 4” with “Figure 5A”.
  • Pag. 8_243: Please replace “Figure 5” with “Figure 6”.

Discussion:

·       Pag. 10_270-272: The wild aquatic birds were the natural host for IAVs, H1-H16 subtypes were present in wild birds; however, some subtypes e.g., H8, H12, H14, and H15 have rarely been detected [32,33].” As previously suggested, you could incorporate further references.

·       Pag. 10_292-294: “IAVs can cross species through reassortment and point mutations to acquire virulence factors required for adaption in a new host, ………..”. Reassortment and point mutations required for IAV adaption in a new host 2 are not necessarily associated with virulence factors.

·       Pag. 10_296-297: “detected” or “were detected”?

·       Pag. 10_306: Please replace “Figure 4” with “Figure 5”.

·       Pag. 10_308: Please replace “Figure 5” with “Figure 6”.

Author Response

Dear Prof.,

We are very grateful for the thoughtful comments made by you. We have carefully addressed these comment and revised our manuscript (the revisions were indicated by tracked changes). The following is the point-to-point rebuttal letter and the attachment is the revised manuscript. We hope meet with approval. let us konw if there is any question about out revision or anything we need to do.

Sincerely,

Zhimin Wan, PhD.

Response to Review 1 comments

In this article the Authors analyzed three novel H12 influenza A viruses (IAVs) detected and isolated (1/3) from Anas spp. in Poyang Lake of China in 2018. Phylogenetic analysis showed that the genome sequences of the three H12 viruses belonged to Eurasian lineage, except for PA genes and one of NP gene, which belonged to North American lineage. The growth kinetics showed that the H12 isolate grew better in A549 than MDCK cells. This H12 isolate cannot efficiently replicate in BALB/c mice, but it can bind to both α-2,6 sialic acid (SA) and α-2,3 SA.

The phylodynamics of H12 viruses, examined by Bayesian phylogeographic analysis, showed these spatial and transmission patterns: two major transmission routes of H12 IAVs were from Asia to Oceania and from Europe to South America, and Anas and Arenaria genera represented the major hosts of the viral transmission.

According to the Authors’ conclusions, the data obtained help to better understand the evolution of H12 IAV and emphasize the need for IAV surveillance activities in wild birds.

Broad comments

This article provides an in-depth scientific analysis aimed to better understand the evolution of H12 IAV in wild birds. Authors performed virus detection and isolation followed by molecular characterization, in vitro growth kinetics and animal study experiments as well as phylogenetic, evolutionary and phylodynamic analyses.

However, in my opinion, the manuscript needs some substantial revisions and minor modifications aimed to improve its clarity.

Please, see “Specific comments” for details.

Specific comments

Abstract:

Point 1. Pag. 1_19: “..... is ubiquitous in wild birds.”. I suggest to modify this sentence by specifying that IAV is widespread in the wild bird reservoir.

Response 1: Thank you for your suggestion, we have modified the sentence in the revised version in line 19.

Point 2. Pag. 1_19-20: I suggest to integrate the sentence “Although hemagglutinin subtypes of H1-H16 are …. as follows “Sixteen hemagglutinin subtypes (from H1 to H16) are ….”

Response 2: Thank you for your suggestion, we have modified the sentence in the revised version in line 19.

Point 3. Pag. 1_20: “very infrequently”: you could use the adverb “infrequently” only; alternatively, could you specify in which continents H12 viruses are more rarely detected?

Response 3: Thank you for your suggestion, we have deleted “very” in the revised version in line 20.

Point 4. Pag. 1_20: I suggest to complete the genus “Anas ….” with the duck species found to be H12 positive “Anas crecca and A. Formosa ….”. Please, see also the comment to the results on Pag..

Response 4: Thank you for your suggestion, we have changed “Anas” into “Anas crecca and Anas formosa” in the revised version in line 21.

Point 5. Pag. 1_26: I suggest to integrate “…. α-2,6 sialic acid (SA) and α-2,3 SA.” with “…. α-2,6 sialic acid (SA) and α-2,3 SA-linked receptors.”

Response 5: Thank you for your suggestion, we have changed “…. α-2,6 sialic acid (SA) and α-2,3 SA.” into“…. α-2,6 sialic acid (SA) and α-2,3 SA-linked receptors.” in the revised version in line 26-27.

Point 6. Pag. 1_27-29: “The results showed that two major transmission routes of H12 IAVs were from Asia to Oceania and from Europe to South America, and Anas and Arenaria were the major hosts of the viral transmission.” Please see comments on Pag. 6_205-207.

Response 6: Thank you for your comment. East Asian-Australasian Flyway is probably responsible for the H12 spreading from Asia to Oceania. For the Europe to South American, the H12 probably spread through an unknown way or overlapped flyway. However, the H12 viruses from South America were clustered with those from America in the MCC tree (Figure 2B), indicating the H12 spread in South America also probably came from North America. Since H12 virus is difficult to be efficiently isolated, some data for H12 might be missing and the sampling bias could affect the robustness of the results. We added the related information in line 337-343 in the revised manuscript.

 Point 7. Pag. 1_29: I suggest to integrate “…., and Anas and Arenaria were…..” as follows ““…., and  Anas and Arenaria genera were …..”

Response 7:  Thank you for your suggestion, we have modified the sentence accordingly in line 29 in the revised version.

Introduction:

Point 8. Pag. 1_40: “…. migratory waterfowl …..”. Did you mean Anseriformes birds, or more generally " aquatic birds" (larger group of birds associated with water habitats)? Please, specify.

Response 8: Thank you for your comment, we changed migratory waterfowl” into " aquatic birds" in the revised version in line 41.

Point 9. Pag. 2_56: “H12 subtype IAV is also waterfowl-associated, but it is infrequently detected in nature; however, it can be reassort with other subtypes [9]”. In addition to [9] , I suggest adding further pertinent references, such as: i) Tang L, Tang W, Ming L, Gu J, Qian K, Li X, Wang T, He G. Characterization of Avian Influenza Virus H10-H12 Subtypes Isolated from Wild Birds in Shanghai, China from 2016 to 2019. Viruses. 2020 Sep 25;12(10):1085. doi: 10.3390/v12101085; ii) Sharshov K, Mine J, Sobolev I, Kurskaya O, Dubovitskiy N, Kabilov M, Alikina T, Nakayama M, Tsunekuni R, Derko A, Prokopyeva E, Alekseev A, Shchelkanov M, Druzyaka A, Gadzhiev A, Uchida Y, Shestopalov A, Saito T. Characterization and Phylodynamics of Reassortant H12Nx Viruses in Northern Eurasia. Microorganisms. 2019 Dec 3;7(12):643. doi: 10.3390/microorganisms7120643.

Response 9: Thank you for your comment and suggestion, we have added these references in revised version in line 58.

Materials and Methods:

Point 10. Pag. 2_62: methods related to the Amino acid analysis of the Three H12 viruses are lacking in this section.

Response 10: Thank you for your comment. The amino acids of the three H12 viruses were examined with DNAStar software. We have added related content in revised version in line 69-79.

Point 11. Pag. 2_64: “A total of 981 cloacal samples were collected from wild birds in Poyang Lake in 2018.”. If possible, could you add some information about the sampling methods and study area (e.g., geographic coordinates, lake size), the samples collected (numbers of wild birds, grouped at least by taxonomic order), and the study period (e.g. months of sampling)?

Response 11: Thank you for your comment and question. We provided the location for Poyang Lake as “113°35’-118°29’ east longitude and 24°29’-30°05’ north latitude” in line 67 in the revised version. Notably, if we provide the detailed information for these 981 samples, there would be too much description. Also, we thought these additional information is out of the scope of this study.

Point 12. Pag. 2_73: “The sequences of H12 IAVs in public databases (NCBI and GISAID) were collected ….”. Please, specify the download period. I suggest adding a sentence about the number of sequences in the obtained database. Moreover, does this database also include data from domestic birds? if so, could you please quantify this sample?

Response 12:  Thank you for your comment. In this study, the sequences of H12 IAVs in public databases (NCBI and GISAID) were collected by October 2021, and total of 377 PB2, 382 PB1, 380 PA, 398 HA, 386 NP, 373 NA, 387 M and 386 NS genes were collected. We added these information in updated manuscript in line 78-79.

Point 13. Pag. 3_106: “Vitro growth kinetics”. I suggest to replace with “In vitro growth kinetics” in the revised manuscript.

Response 13: Thank you for your suggestion. We have changed “Vitro growth kinetics” into “In vitro growth kinetics” in the revised version in line 111. 

Results:

Point 14. Pag.3_141: Due to recent updates of scientific and common names of wild anatids, I suggest specifying the scientific names of Eurasian teal (Anas crecca) https://birdsoftheworld.org/bow/species/gnwtea/cur/introduction and Baikal teal (Anas formosa, more recently classified as Sibirionetta formosa).https://birdsoftheworld.org/bow/species/baitea/cur/introduction.

Response 14: Thank you for your suggestion, we have modified related information in revised version in line 152-155.

Point 15. Pag.4_147-149: “however, the dominant subtypes of H12 vruses circulating in China were H12N2 (5/12), H12N1 (2/12) and H12N8 (2/12) (data not shown)”. In general, the data obtained from China should be further discussed (e.g., origin of H12 viruses and interactions between wild and domestic birds). Please, see also: Liu J, Zhou L, Luo Z, Zou Y, Lv J, Wang T. Genomic characteristics and phylogenetic analysis of the first H12N2 influenza A virus identified from wild birds, China. Acta Virol. 2020;64(1):104-110. doi: 10.4149/av_2020_113.

Response 15: Thank you for your suggestion. So far, only 12 H12 viruses were reported in China. They were mainly detected from Anas (8/12). The common teal or Eurasian green-winged teal (6/12) is a common and widespread duck which breeds temperate Eurosiberia and migrates south in winter. We added the related information in the revised version line 158-162.

Point 16. Pag.4_149: “Host genus analysis showed that dominant hosts ……”.  Actually, this analysis mainly includes hosts genera, but also some different IAV origins are reported (shorebirds, guineafowl, environment, ducks). Please, specify this.

Response 16: Thank you for your comment. We combined the duck into Anas in Figure 2B. For the other hosts (shorebirds, guineafowl, environment), we don’t have further information that can classify these hosts into a genus. 

Point 17. Pag. 4_152-160: Figure 1 is very interesting and useful. However, the small font size and poor resolution hinder the consultation of the legend. I believe that figures 1A, 1B, and 1C should be enlarged and layout differently. Also, I wonder if your three H12 viruses are included in Figure 1. Finally, I would like to point out that the genus Sibirionetta included in the Figure 1 dataset is (very likely) a Baikal teal.

Response 17: Thank you for your suggestion. We have remade Figure 1, and the three H12 viruses in this study were not in Figure 1.

Point 18. Pag. 4_163-164: “(377 PB2, 382 PB1, 380 PA, 398 163 HA, 386 NP, 373 NA, 387 M and 386 NS)”. Please integrate as follows “(377 PB2, 382 PB1, 380 PA, 398 163 HA, 386 NP, 373 NA, 387 M and 386 NS genes).

Response 18: Thank you for your suggestion, we changed “377 PB2, 382 PB1, 380 PA, 398 163 HA, 386 NP, 373 NA, 387 M and 386 NS” into 377 PB2, 382 PB1, 380 PA, 398 163 HA, 386 NP, 373 NA, 387 M and 386 NS genes” in line 172-173 of the revised version.

Point 19. Pp. 5 and 6: As pointed out for Figure 1, the legend of Figure 2B and, in general, Figures 3A through 3F are also difficult to understand.

Response 19: Thank you for your comment, we have updated the legend of Figure 2 and Figure 3 in the revised version.

Point 20. Pag. 6_205-207: Two transmission routes are decisively supports, including migration from Asia to Oceania (BF=29353.79) and Europe to South America (BF=4889.581).” Although statistically supported, the transmission pattern of H12 virus from Europe to South America, also shown in Figure 4B, does not find an explanation in the birds' natural migratory routes. This result should be interpreted/discussed.

Response 20: Thank you for your comment. East Asian-Australasian Flyway is probably responsible for the H12 spreading from Asia to Oceania. For the Europe to South America, the H12 probably spread through an unknown way or overlapped flyway. However, the H12 viruses from South America were clustered with those from America in the MCC tree (Figure 2B), indicating the H12 spread in South America also probably came from North America. Since H12 virus is difficult to be isolated, the H12 data might be missing and the sampling bias could affect the robustness of the results. We added these information in the revised version in line 337-343.

Point 21. Pag. 7_ Figure 1A: Please check the color of the arrows in both the figure and the legend.

Response 21: Thank you for your suggestion. We checked it. It seems ok.

Point 22. Pp. 7 and 8_Table 1: Please check virus strains for E627K and L473V mutations.

Response 22: Thank you for your comment, this is our mistake, we have fixed it in Table 1 and line 256 in the revised version.

Point 23. Pag. 8: 235-236: Please, amend “106.366 235 TCID50/ml”.

Response 23: Thank you for your suggestion, this is our mistake, we have fixed it in the revised version.

Point 24. Pag. 8_236: Please replace “Figure 4” with “Figure 5A”.

Response 24: Thank you for your comment. This is our mistake, we have fixed it in the revised version.

Point 25 Pag. 8_243: Please replace “Figure 5” with “Figure 6”.

Response 25: Thank you for your comment. This is our mistake, we have fixed it in the revised version.

Discussion:

Point 26. Pag. 10_270-272: “The wild aquatic birds were the natural host for IAVs, H1-H16 subtypes were present in wild birds; however, some subtypes e.g., H8, H12, H14, and H15 have rarely been detected [32,33].” As previously suggested, you could incorporate further references.

Response 26: Thank you for your suggestion. We cited these related references in line 305 of the revised version.

Point 27. Pag. 10_292-294: “IAVs can cross species through reassortment and point mutations to acquire virulence factors required for adaption in a new host, ………..”. Reassortment and point mutations required for IAV adaption in a new host 2 are not necessarily associated with virulence factors.

Response 27: Thank you for your comment. We changed this sentence into “IAVs can cross species through reassortment and point mutations to adaption to a new host.” in revised manuscript in line 329-330.

Point 28. Pag. 10_296-297: “detected” or “were detected”?

Response 28: Thank you for your question, it is “were detected, we have fixed it in line33 of the revised manuscript.

 Point 29. Pag. 10_306: Please replace “Figure 4” with “Figure 5”.

Response 29: Thank you for your comment. This is our mistake, we have fixed it in the revised manuscript.

Point 30.  Pag. 10_308: Please replace “Figure 5” with “Figure 6”.

Response 30: Thank you for your comment. This is our mistake, we have fixed it in the revised manuscript.

Reviewer 2 Report

Wan et al. detected and characterized three H12 subtype avian influenza A viruses (IAV) in wild birds in Poyang Lake. Phylogenetic analysis revealed that the three H12 IAVs belong to Eurasian lineage, expect for PA genes and one of NP gene, which belong to North American lineage. Authors also evaluated the biological characterizations of one H12 isolate in vitro and in vivo. Results showed that although this H12 isolate cannot efficiently replicate in mice, it can efficiently replicate in MDCK and A549 cells and bind both human-like and avian-like receptors. Moreover, authors detects that two major transmission routes of H12 IAVs were from Asia to Oceania and from Europe to South America, and Anas and Arenaria are the major hosts of the viral transmission. Over all, the conclusions of the manuscript are supported by the data presented. However, some comments and suggestions that the authors should consider to hopefully improve the manuscript.

1. Please indicate whether other subtypes of IAV have been detected or isolated in these 981 cloacal samples tested.

2. Please indicate why only B-Teal/354 of three positive samples was efficiently isolated.

3. The Figure 1B showed that most H12 viruses were detected in North America, please indicate the phylogeography in North America.

4. Please add corresponding citations into Table 1.

5. A statistical analysis was performed in Figure 5b, please add the corresponding methods for the statistical assay in the manuscript.

6. Are there any differences in the plaque phenotype of H12 isolate in MDCK and A549 cells?

7. The Figure 2 should be zoom in.

8. Table 1-Please change “amanradine and remantadine” to “Amantadine and Rimantadine”.

9. Please, keep consistency with writing of host (Anas and Arenaria) of H12 IAV (Italics or normal).

10. Line 182-Please change “HA gens” to “HA genes”.

11. Line 197-Please change “these study” to “this study”.

12. Line 213- Please change “supports” to “supported”

Author Response

Dear Prof., 

We are very grateful for the thoughtful comments made by you. We have carefully addressed these comment and revised our manuscript (the revisions were indicated by tracked changes). The following is the point-to-point rebuttal letter and the attachment is the revised manuscript. We hope meet with approval. Please let us konw if there is any question about out revision or anything we need to do.

Sincerely,

Zhimin Wan, PhD.

Response to Review 1 comments

In this article the Authors analyzed three novel H12 influenza A viruses (IAVs) detected and isolated (1/3) from Anas spp. in Poyang Lake of China in 2018. Phylogenetic analysis showed that the genome sequences of the three H12 viruses belonged to Eurasian lineage, except for PA genes and one of NP gene, which belonged to North American lineage. The growth kinetics showed that the H12 isolate grew better in A549 than MDCK cells. This H12 isolate cannot efficiently replicate in BALB/c mice, but it can bind to both α-2,6 sialic acid (SA) and α-2,3 SA.

The phylodynamics of H12 viruses, examined by Bayesian phylogeographic analysis, showed these spatial and transmission patterns: two major transmission routes of H12 IAVs were from Asia to Oceania and from Europe to South America, and Anas and Arenaria genera represented the major hosts of the viral transmission.

According to the Authors’ conclusions, the data obtained help to better understand the evolution of H12 IAV and emphasize the need for IAV surveillance activities in wild birds.

Broad comments

This article provides an in-depth scientific analysis aimed to better understand the evolution of H12 IAV in wild birds. Authors performed virus detection and isolation followed by molecular characterization, in vitro growth kinetics and animal study experiments as well as phylogenetic, evolutionary and phylodynamic analyses.

However, in my opinion, the manuscript needs some substantial revisions and minor modifications aimed to improve its clarity.

Please, see “Specific comments” for details.

Specific comments

Abstract:

Point 1. Pag. 1_19: “..... is ubiquitous in wild birds.”. I suggest to modify this sentence by specifying that IAV is widespread in the wild bird reservoir.

Response 1: Thank you for your suggestion, we have modified the sentence in the revised version in line 19.

Point 2. Pag. 1_19-20: I suggest to integrate the sentence “Although hemagglutinin subtypes of H1-H16 are …. as follows “Sixteen hemagglutinin subtypes (from H1 to H16) are ….”

Response 2: Thank you for your suggestion, we have modified the sentence in the revised version in line 19.

Point 3. Pag. 1_20: “very infrequently”: you could use the adverb “infrequently” only; alternatively, could you specify in which continents H12 viruses are more rarely detected?

Response 3: Thank you for your suggestion, we have deleted “very” in the revised version in line 20.

Point 4. Pag. 1_20: I suggest to complete the genus “Anas ….” with the duck species found to be H12 positive “Anas crecca and A. Formosa ….”. Please, see also the comment to the results on Pag..

Response 4: Thank you for your suggestion, we have changed “Anas” into “Anas crecca and Anas formosa” in the revised version in line 21.

Point 5. Pag. 1_26: I suggest to integrate “…. α-2,6 sialic acid (SA) and α-2,3 SA.” with “…. α-2,6 sialic acid (SA) and α-2,3 SA-linked receptors.”

Response 5: Thank you for your suggestion, we have changed “…. α-2,6 sialic acid (SA) and α-2,3 SA.” into“…. α-2,6 sialic acid (SA) and α-2,3 SA-linked receptors.” in the revised version in line 26-27.

Point 6. Pag. 1_27-29: “The results showed that two major transmission routes of H12 IAVs were from Asia to Oceania and from Europe to South America, and Anas and Arenaria were the major hosts of the viral transmission.” Please see comments on Pag. 6_205-207.

Response 6: Thank you for your comment. East Asian-Australasian Flyway is probably responsible for the H12 spreading from Asia to Oceania. For the Europe to South American, the H12 probably spread through an unknown way or overlapped flyway. However, the H12 viruses from South America were clustered with those from America in the MCC tree (Figure 2B), indicating the H12 spread in South America also probably came from North America. Since H12 virus is difficult to be efficiently isolated, some data for H12 might be missing and the sampling bias could affect the robustness of the results. We added the related information in line 337-343 in the revised manuscript.

 Point 7. Pag. 1_29: I suggest to integrate “…., and Anas and Arenaria were…..” as follows ““…., and  Anas and Arenaria genera were …..”

Response 7:  Thank you for your suggestion, we have modified the sentence accordingly in line 29 in the revised version.

Introduction:

Point 8. Pag. 1_40: “…. migratory waterfowl …..”. Did you mean Anseriformes birds, or more generally " aquatic birds" (larger group of birds associated with water habitats)? Please, specify.

Response 8: Thank you for your comment, we changed migratory waterfowl” into " aquatic birds" in the revised version in line 41.

Point 9. Pag. 2_56: “H12 subtype IAV is also waterfowl-associated, but it is infrequently detected in nature; however, it can be reassort with other subtypes [9]”. In addition to [9] , I suggest adding further pertinent references, such as: i) Tang L, Tang W, Ming L, Gu J, Qian K, Li X, Wang T, He G. Characterization of Avian Influenza Virus H10-H12 Subtypes Isolated from Wild Birds in Shanghai, China from 2016 to 2019. Viruses. 2020 Sep 25;12(10):1085. doi: 10.3390/v12101085; ii) Sharshov K, Mine J, Sobolev I, Kurskaya O, Dubovitskiy N, Kabilov M, Alikina T, Nakayama M, Tsunekuni R, Derko A, Prokopyeva E, Alekseev A, Shchelkanov M, Druzyaka A, Gadzhiev A, Uchida Y, Shestopalov A, Saito T. Characterization and Phylodynamics of Reassortant H12Nx Viruses in Northern Eurasia. Microorganisms. 2019 Dec 3;7(12):643. doi: 10.3390/microorganisms7120643.

Response 9: Thank you for your comment and suggestion, we have added these references in revised version in line 58.

Materials and Methods:

Point 10. Pag. 2_62: methods related to the Amino acid analysis of the Three H12 viruses are lacking in this section.

Response 10: Thank you for your comment. The amino acids of the three H12 viruses were examined with DNAStar software. We have added related content in revised version in line 69-79.

Point 11. Pag. 2_64: “A total of 981 cloacal samples were collected from wild birds in Poyang Lake in 2018.”. If possible, could you add some information about the sampling methods and study area (e.g., geographic coordinates, lake size), the samples collected (numbers of wild birds, grouped at least by taxonomic order), and the study period (e.g. months of sampling)?

Response 11: Thank you for your comment and question. We provided the location for Poyang Lake as “113°35’-118°29’ east longitude and 24°29’-30°05’ north latitude” in line 67 in the revised version. Notably, if we provide the detailed information for these 981 samples, there would be too much description. Also, we thought these additional information is out of the scope of this study.

Point 12. Pag. 2_73: “The sequences of H12 IAVs in public databases (NCBI and GISAID) were collected ….”. Please, specify the download period. I suggest adding a sentence about the number of sequences in the obtained database. Moreover, does this database also include data from domestic birds? if so, could you please quantify this sample?

Response 12:  Thank you for your comment. In this study, the sequences of H12 IAVs in public databases (NCBI and GISAID) were collected by October 2021, and total of 377 PB2, 382 PB1, 380 PA, 398 HA, 386 NP, 373 NA, 387 M and 386 NS genes were collected. We added these information in updated manuscript in line 78-79.

Point 13. Pag. 3_106: “Vitro growth kinetics”. I suggest to replace with “In vitro growth kinetics” in the revised manuscript.

Response 13: Thank you for your suggestion. We have changed “Vitro growth kinetics” into “In vitro growth kinetics” in the revised version in line 111. 

Results:

Point 14. Pag.3_141: Due to recent updates of scientific and common names of wild anatids, I suggest specifying the scientific names of Eurasian teal (Anas crecca) https://birdsoftheworld.org/bow/species/gnwtea/cur/introduction and Baikal teal (Anas formosa, more recently classified as Sibirionetta formosa).https://birdsoftheworld.org/bow/species/baitea/cur/introduction.

Response 14: Thank you for your suggestion, we have modified related information in revised version in line 152-155.

Point 15. Pag.4_147-149: “however, the dominant subtypes of H12 vruses circulating in China were H12N2 (5/12), H12N1 (2/12) and H12N8 (2/12) (data not shown)”. In general, the data obtained from China should be further discussed (e.g., origin of H12 viruses and interactions between wild and domestic birds). Please, see also: Liu J, Zhou L, Luo Z, Zou Y, Lv J, Wang T. Genomic characteristics and phylogenetic analysis of the first H12N2 influenza A virus identified from wild birds, China. Acta Virol. 2020;64(1):104-110. doi: 10.4149/av_2020_113.

Response 15: Thank you for your suggestion. So far, only 12 H12 viruses were reported in China. They were mainly detected from Anas (8/12). The common teal or Eurasian green-winged teal (6/12) is a common and widespread duck which breeds temperate Eurosiberia and migrates south in winter. We added the related information in the revised version line 158-162.

Point 16. Pag.4_149: “Host genus analysis showed that dominant hosts ……”.  Actually, this analysis mainly includes hosts genera, but also some different IAV origins are reported (shorebirds, guineafowl, environment, ducks). Please, specify this.

Response 16: Thank you for your comment. We combined the duck into Anas in Figure 2B. For the other hosts (shorebirds, guineafowl, environment), we don’t have further information that can classify these hosts into a genus. 

Point 17. Pag. 4_152-160: Figure 1 is very interesting and useful. However, the small font size and poor resolution hinder the consultation of the legend. I believe that figures 1A, 1B, and 1C should be enlarged and layout differently. Also, I wonder if your three H12 viruses are included in Figure 1. Finally, I would like to point out that the genus Sibirionetta included in the Figure 1 dataset is (very likely) a Baikal teal.

Response 17: Thank you for your suggestion. We have remade Figure 1, and the three H12 viruses in this study were not in Figure 1.

Point 18. Pag. 4_163-164: “(377 PB2, 382 PB1, 380 PA, 398 163 HA, 386 NP, 373 NA, 387 M and 386 NS)”. Please integrate as follows “(377 PB2, 382 PB1, 380 PA, 398 163 HA, 386 NP, 373 NA, 387 M and 386 NS genes).

Response 18: Thank you for your suggestion, we changed “377 PB2, 382 PB1, 380 PA, 398 163 HA, 386 NP, 373 NA, 387 M and 386 NS” into 377 PB2, 382 PB1, 380 PA, 398 163 HA, 386 NP, 373 NA, 387 M and 386 NS genes” in line 172-173 of the revised version.

Point 19. Pp. 5 and 6: As pointed out for Figure 1, the legend of Figure 2B and, in general, Figures 3A through 3F are also difficult to understand.

Response 19: Thank you for your comment, we have updated the legend of Figure 2 and Figure 3 in the revised version.

Point 20. Pag. 6_205-207: Two transmission routes are decisively supports, including migration from Asia to Oceania (BF=29353.79) and Europe to South America (BF=4889.581).” Although statistically supported, the transmission pattern of H12 virus from Europe to South America, also shown in Figure 4B, does not find an explanation in the birds' natural migratory routes. This result should be interpreted/discussed.

Response 20: Thank you for your comment. East Asian-Australasian Flyway is probably responsible for the H12 spreading from Asia to Oceania. For the Europe to South America, the H12 probably spread through an unknown way or overlapped flyway. However, the H12 viruses from South America were clustered with those from America in the MCC tree (Figure 2B), indicating the H12 spread in South America also probably came from North America. Since H12 virus is difficult to be isolated, the H12 data might be missing and the sampling bias could affect the robustness of the results. We added these information in the revised version in line 337-343.

Point 21. Pag. 7_ Figure 1A: Please check the color of the arrows in both the figure and the legend.

Response 21: Thank you for your suggestion. We checked it. It seems ok.

Point 22. Pp. 7 and 8_Table 1: Please check virus strains for E627K and L473V mutations.

Response 22: Thank you for your comment, this is our mistake, we have fixed it in Table 1 and line 256 in the revised version.

Point 23. Pag. 8: 235-236: Please, amend “106.366 235 TCID50/ml”.

Response 23: Thank you for your suggestion, this is our mistake, we have fixed it in the revised version.

Point 24. Pag. 8_236: Please replace “Figure 4” with “Figure 5A”.

Response 24: Thank you for your comment. This is our mistake, we have fixed it in the revised version.

Point 25 Pag. 8_243: Please replace “Figure 5” with “Figure 6”.

Response 25: Thank you for your comment. This is our mistake, we have fixed it in the revised version.

Discussion:

Point 26. Pag. 10_270-272: “The wild aquatic birds were the natural host for IAVs, H1-H16 subtypes were present in wild birds; however, some subtypes e.g., H8, H12, H14, and H15 have rarely been detected [32,33].” As previously suggested, you could incorporate further references.

Response 26: Thank you for your suggestion. We cited these related references in line 305 of the revised version.

Point 27. Pag. 10_292-294: “IAVs can cross species through reassortment and point mutations to acquire virulence factors required for adaption in a new host, ………..”. Reassortment and point mutations required for IAV adaption in a new host 2 are not necessarily associated with virulence factors.

Response 27: Thank you for your comment. We changed this sentence into “IAVs can cross species through reassortment and point mutations to adaption to a new host.” in revised manuscript in line 329-330.

Point 28. Pag. 10_296-297: “detected” or “were detected”?

Response 28: Thank you for your question, it is “were detected, we have fixed it in line33 of the revised manuscript.

 Point 29. Pag. 10_306: Please replace “Figure 4” with “Figure 5”.

Response 29: Thank you for your comment. This is our mistake, we have fixed it in the revised manuscript.

Point 30.  Pag. 10_308: Please replace “Figure 5” with “Figure 6”.

Response 30: Thank you for your comment. This is our mistake, we have fixed it in the revised manuscript.

Round 2

Reviewer 1 Report

The submitted manuscript “Phylogeography and biological characterizations of H12 influenza A viruses” (viruses-1931083-peer-review-v2) has been thoroughly revised by the Authors. All suggested corrections and changes have been made.

I suggest a minor spell check for the English language, especially for the additions to the text in the revised version (related for example to point 15 of the rebuttal letter).

Moreover, I recommend being careful to avoid distorting the images by enlarging them.

In my opinion, the manuscript may be accepted after a few minor revisions.

Please, see “Specific comments” for details:

Results:

Point 14. Pag.3_141: Due to recent updates of scientific and common names of wild anatids, I suggest specifying the scientific names of Eurasian teal (Anas crecca) https://birdsoftheworld.org/bow/species/gnwtea/cur/introduction and Baikal teal (Anas formosa, more recently classified as Sibirionetta formosa).https://birdsoftheworld.org/bow/species/baitea/cur/introduction.

Authors’ response 14: Thank you for your suggestion, we have modified related information in revised version in line 152-155.

peer-review-v2:

As correctly stated in the full name of H12 strains, the duck's scientific name (italicized) requires that the first letter of the genus name be capitalized and the species name be lowercase.

Therefore, I suggest to replace throughout the manuscript, including Figure 3, the three strain abbreviations A. Crecca/135, A. Formosa/354 and A. Crecca/421 as follows: A. crecca/135, A. formosa/354 and A. crecca/421.

__________________________________________________________________________

Point 16. Pag.4_149: “Host genus analysis showed that dominant hosts ……”.  Actually, this analysis mainly includes hosts genera, but also some different IAV origins are reported (shorebirds, guineafowl, environment, ducks). Please, specify this.

Authors’ response 16: Thank you for your comment. We combined the duck into Anas in Figure 2B. For the other hosts (shorebirds, guineafowl, environment), we don’t have further information that can classify these hosts into a genus

peer-review-v2

I think I did not explain myself well. I meant that since it is not possible to indicate the genus of all birds one could replace in Figure 1A and Figure 2B “Host Genus” with “Host” and specify in the legend of Figure 1 that " (A) The map of the subtypes, hosts (including genus/type of birds or environment), and continents ......".

Unless you have additional information, it would also be better not to include the 20 unspecified ducks in the genus Anas as they may belong to other genera (e.g., Mareca, Spatula, Aythya, .....). In addition, these 20 ducks are actually still in Figure 1.

__________________________________________________________________________

Point 21. Pag 7_Figure 1A: Please check the color of the arrows in both the figure and the legend.

Response 21: Thank you for your suggestion. We checked it. It seems ok.

peer-review-v2

I also seem to see gray arrows.

Author Response

Dear Prof.,

We are very grateful for your thoughtful comments. We have carefully addressed these comment and revised our manuscript. We hope now it suitable for publishing. The following is the rebuttal letter. Pleace check it.

Sincerely,

Zhimin Wan

The submitted manuscript “Phylogeography and biological characterizations of H12 influenza A viruses” (viruses-1931083-peer-review-v2) has been thoroughly revised by the Authors. All suggested corrections and changes have been made.

I suggest a minor spell check for the English language, especially for the additions to the text in the revised version (related for example to point 15 of the rebuttal letter).

Moreover, I recommend being careful to avoid distorting the images by enlarging them.

In my opinion, the manuscript may be accepted after a few minor revisions.

Please, see “Specific comments” for details:

Results:

Point 14. Pag.3_141: Due to recent updates of scientific and common names of wild anatids, I suggest specifying the scientific names of Eurasian teal (Anas crecca) https://birdsoftheworld.org/bow/species/gnwtea/cur/introduction and Baikal teal (Anas formosa, more recently classified as Sibirionetta formosa).https://birdsoftheworld.org/bow/species/baitea/cur/introduction.

Authors’ response 14: Thank you for your suggestion, we have modified related information in revised version in line 152-155.

peer-review-v2:

As correctly stated in the full name of H12 strains, the duck's scientific name (italicized) requires that the first letter of the genus name be capitalized and the species name be lowercase.

Therefore, I suggest to replace throughout the manuscript, including Figure 3, the three strain abbreviations A. Crecca/135, A. Formosa/354 and A. Crecca/421 as follows: A. crecca/135, A. formosa/354 and A. crecca/421.

Response:Thank you for your suggestion, we have changed “ A. Crecca/135, A. Formosa/354 and A. Crecca/421” to “A. crecca/135, A. formosa/354 and A. crecca/421” throughout the manuscript in the revised manuscript.

_________________________________________________________________

Point 16. Pag.4_149: “Host genus analysis showed that dominant hosts ……”.  Actually, this analysis mainly includes hosts genera, but also some different IAV origins are reported (shorebirds, guineafowl, environment, ducks). Please, specify this.

Authors’ response 16: Thank you for your comment. We combined the duck into Anas in Figure 2B. For the other hosts (shorebirds, guineafowl, environment), we don’t have further information that can classify these hosts into a genus

peer-review-v2

I think I did not explain myself well. I meant that since it is not possible to indicate the genus of all birds one could replace in Figure 1A and Figure 2B “Host Genus” with “Host” and specify in the legend of Figure 1 that " (A) The map of the subtypes, hosts (including genus/type of birds or environment), and continents ......".

Unless you have additional information, it would also be better not to include the 20 unspecified ducks in the genus Anas as they may belong to other genera (e.g., Mareca, SpatulaAythya, .....). In addition, these 20 ducks are actually still in Figure 1.

Response: Thank you for your comment. We have replaced “Host Genus” with Figure 1A and Figure 2B and remade Figure 2B in the revised manuscript .

__________________________________________________________________________

Point 21. Pag 7_Figure 1A: Please check the color of the arrows in both the figure and the legend.

Response 21: Thank you for your suggestion. We checked it. It seems ok.

peer-review-v2

I also seem to see gray arrows.

Response: Thank you for your comment. We remade Figure 1A, and use different colors to represent different subtypes of H12 viruses in the revised manuscript.

We also checked our manuscript carefully, and improved the manuscript

, and undated the Figures. We hope it now suitable for publishing.
